# An annotation-free whole-slide training approach to pathological classification of lung cancer types using deep learning

Chi-Long Chen [1,2,3,9], Chi-Chung Chen [4,9], Wei-Hsiang Yu [4], Szu-Hua Chen [4], Yu-Chan Chang [5], Tai-I Hsu[6], Michael Hsiao [6], Chao-Yuan Yeh [4✉] & Cheng-Yu Chen[7,8✉]

Deep learning for digital pathology is hindered by the extremely high spatial resolution of whole-slide images (WSIs). Most studies have employed patch-based methods, which often require detailed annotation of image patches. This typically involves laborious free-hand contouring on WSIs. To alleviate the burden of such contouring and obtain benefits from scaling up training with numerous WSIs, we develop a method for training neural networks on entire WSIs using only slide-level diagnoses. Our method leverages the unified memory mechanism to overcome the memory constraint of compute accelerators. Experiments conducted on a data set of 9662 lung cancer WSIs reveal that the proposed method achieves areas under the receiver operating characteristic curve of 0.9594 and 0.9414 for adeno-carcinoma and squamous cell carcinoma classification on the testing set, respectively. Furthermore, the method demonstrates higher classification performance than multiple-instance learning as well as strong localization results for small lesions through class activation mapping.

---

[1] Department of Pathology, School of Medicine, College of Medicine, Taipei Medical University, Taipei, Taiwan. [2] Department of Pathology, Taipei Medical University Hospital, Taipei, Taiwan. [3] Research Center for Artificial Intelligence in Medicine, Taipei Medical University, Taipei, Taiwan. [4] aetherAI Co., Ltd., Taipei, Taiwan. [5] Department of Biomedical Imaging and Radiological Sciences, National Yang-Ming University, Taipei, Taiwan. [6] Genomics Research Center, Academia Sinica, Taipei, Taiwan. [7] Department of Radiology, School of Medicine, College of Medicine, Taipei Medical University, Taipei, Taiwan. [8] Department of Radiology, Taipei Medical University Hospital, Taipei, Taiwan. [9] These authors contributed equally: Chi-Long Chen, Chi-Chung Chen. ✉email: joeyeh@aetherai.com; sandychen@tmu.edu.tw

In recent decades, lung cancer has been among the most frequently diagnosed cancers and the leading cause of cancer-related mortality worldwide, including in Taiwan[1]. Nonsmall-cell lung cancer (NSCLC) accounts for ~85% of newly diagnosed lung cancer cases, with two major histological types: adenocarcinoma and squamous cell carcinoma, accounting for nearly 50% and 30% of NSCLC, respectively[2]. Invasive adenocarcinoma of the lungs is a malignant epithelial tumor with five major patterns: lepidic, acinar, papillary, micropapillary, and solid. Squamous cell carcinoma is a malignant epithelial tumor with squamous differentiation and/or keratinization. Proper pathologic diagnosis can be challenging in many cases because morphological differences among lung cancer types are subtle. Examples of the pathological features of adenocarcinoma and squamous cell carcinoma are presented in Fig. 1.

Deep neural networks (DNNs), especially convolutional neural networks (CNNs), have become the dominant method for image recognition; in 2012, their performance surpassed most traditional image analysis algorithms in the ImageNet Large Scale Visual Recognition Challenge[3–6]. In medical fields, deep learning algorithms have also been demonstrated to achieve human-level performance on several tasks, including tumor identification and segmentation in computed tomography or magnetic resonance imaging[7,8], cardiovascular risk assessment using color fundus images[9], and pneumonia detection in chest X-rays[10]. However, the analysis of digital whole-slide images (WSIs) remains challenging because of their extremely high spatial resolution compared with other medical imaging modalities.

Restricted by computing limitations, most histopathology studies have used a two-stage patch-based workflow: a patch-level CNN is trained using patches cropped from a WSI, followed by a slide-level algorithm being trained on features extracted by the patch-level model to reveal the final diagnosis. These patch-based methods have yielded successful results in cancer identification[11–19], cancer type classification[14,20], cancer metastasis detection[13,16–18], and prognosis analysis[21,22]. However, such methods require experienced pathologists to perform substantial annotation.

To leverage slide-level labels directly, multiple-instance learning (MIL)[19,23,24] follows the same two-stage workflow as the traditional method while organizing the training procedure differently. In MIL for slide-level cancer classification, if patches with the highest scores ($k$ patches that are most likely to be cancerous) on the slide are identified as carcinoma, the slide should be classified as cancer; otherwise, when patches with top scores are normal, the slide is classified as benign. By using the slide-level ground truth as weak supervision, MIL successfully reduces the annotation burden; however, recent studies have indicated that even state-of-the-art weak supervision methods still cannot achieve the average performance of strong supervision methods in most image recognition tasks such as object detection, semantic segmentation, and instance segmentation[25–27]. Unlike the MIL method, which selects the top $k$ patches as slide representatives to train models iteratively, the streaming CNN[28,29] proposed by Pinckaers et al. incorporates patching into a back-propagation algorithm to achieve end-to-end training of large images. Specifically, the streaming CNN collects and updates loss gradients of patches of a WSI during training with a specially designed update schedule; hence, it can retain all image information with limited computing resources. However, patching feature maps during training disrupts some operations that require all of the feature maps' information, such as the most commonly used batch normalization layer; thus, careful model design and tuning are required.

In this study, we developed a whole-slide training method that incorporates the unified memory (UM) mechanism and several GPU memory optimization techniques to train standard CNNs with extremely large image inputs without modification in either training pipelines or model architectures. The results of experiments revealed that the proposed method can be directly applied to WSI classification and outperforms the MIL method. The study's contributions are summarized as follows: (1) We propose a training approach to train CNNs on WSIs using slide-level labels without dividing the input image or feature maps into patches. (2) Our method has superior performance, achieving area under the receiver operating characteristic curve (AUC) scores of 0.9594 and 0.9414 for adenocarcinoma and squamous cell carcinoma classification, respectively. (3) Critical regions of our model highlighted by the class activation map (CAM)[30] technique reveal a high correspondence to cancerous regions identified by pathologists.

## Results

Unless otherwise specified, experiments were conducted with slides scanned at ×20 magnification, which were downscaled to ×4 magnification (i.e., resized to 0.2 times the original size) to train both the whole-slide and MIL models. After downscaling to

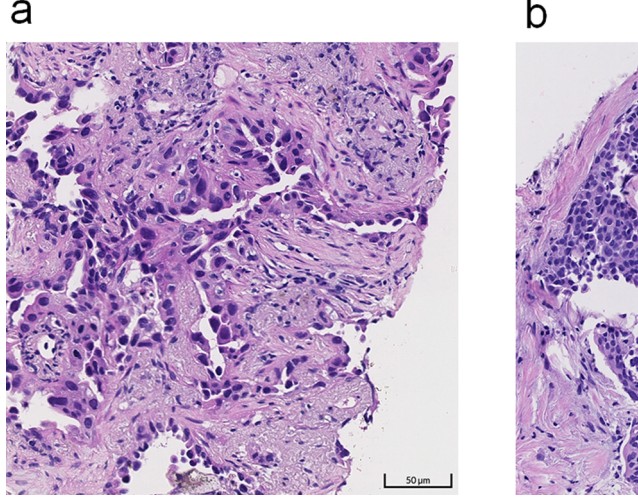

**Fig. 1 Examples of pathological images of major lung cancers.** Pathological images of adenocarcinoma and squamous cell carcinoma are displayed in **a** and **b**, respectively.

**Table 1 Model performances of various MIL methods and our proposed whole-slide training method.**

| Method | AUCs of ADC | AUCs of SqCC |
|---|---|---|
| Results on the standard testing data set ($n = 1397$) | | |
| MIL ($k = 1$) | 0.8922 (0.8759–0.9084) | 0.8513 (0.8262–0.8765) |
| MIL ($k = 3$) | 0.9188 (0.9052–0.9324) | 0.9032 (0.8825–0.9240) |
| MIL ($k = 5$) | 0.9124 (0.8977–0.9270) | 0.8912 (0.8682–0.9142) |
| EM-CNN-LR | 0.7700 (0.7457–0.7944) | 0.8247 (0.7960–0.8534) |
| EM-CNN-SVM | 0.7671 (0.7427–0.7916) | 0.7726 (0.7358–0.8094) |
| CNN-MaxFeat-based RF ($k = 3$) | 0.9345 (0.9223–0.9467) | 0.9071 (0.8855–0.9286) |
| MIL-RNN ($k = 3$) | 0.9310 (0.9185–0.9435) | 0.9239 (0.9047–0.9431) |
| Whole-slide training (GAP) | 0.6506 (0.6213–0.6798) | 0.5597 (0.5176–0.6018) |
| Whole-slide training (GMP) | 0.9594 (0.9500–0.9689) | 0.9414 (0.9234–0.9593) |
| Results on the small lesion data set ($n = 476$) | | |
| CNN-MaxFeat-based RF ($k = 3$) | 0.8823 (0.8129–0.9517) | 0.8727 (0.8111–0.9343) |
| MIL-RNN ($k = 3$) | 0.8681 (0.7902–0.9460) | 0.9100 (0.8625–0.9574) |
| Whole-slide training (GMP) | 0.9384 (0.8849–0.9919) | 0.9202 (0.8740–0.9664) |
| Results on TCGA-diagnostic data set ($n = 1044$) | | |
| CNN-MaxFeat-based RF ($k = 3$) | 0.8319 (0.8062–0.8576) | 0.8447 (0.8219–0.8674) |
| MIL-RNN ($k = 3$) | 0.8601 (0.8374–0.8827) | 0.8752 (0.8550–0.8955) |
| Whole-slide training (GMP) | 0.8950 (0.8764–0.9137) | 0.8990 (0.8811–0.9169) |
| Results on TCGA-tissue data set ($n = 2167$) | | |
| CNN-MaxFeat-based RF ($k = 3$) | 0.6933 (0.6714–0.7153) | 0.6609 (0.6381–0.6836) |
| MIL-RNN ($k = 3$) | 0.7096 (0.6880–0.7312) | 0.6312 (0.6079–0.6546) |
| Whole-slide Training (GMP) | 0.7413 (0.7207–0.7619) | 0.7348 (0.7141–0.7555) |

Performances are measured by area under the receiver operating characteristic curves (AUCs) of classifying lung adenocarcinoma (ADC) and squamous cell carcinoma (SqCC) on the standard testing data set ($n = 1397$), the small lesion data set ($n = 476$), the TCGA-diagnostic data set ($n = 1044$), and the TCGA-tissue data set ($n = 2167$). All measurements were taken from distinct samples. The 95% confidence intervals were estimated by Delong's method.

×4 magnification, most tissue on the slides could be included with a height and width of 21,500 pixels. The downsampled images were then padded to 21,500 × 21,500 to ensure identical size. We used ResNet-50[3] with fixup initialization[31] for all experiments. The models were trained as a ternary classifier for adenocarcinoma, squamous cell carcinoma, or non-cancer. Models were trained on 5606 slides and evaluated on 1397 slides collected from Taipei Medical University Hospital (TMUH), Taipei Municipal Wanfang Hospital (WFH), and Taipei Medical University Shuang-Ho Hospital (SHH). Finally, model performances for lung cancer type classification in lung specimens were measured using the AUC.

**MIL model performance**. We conducted experiments using both standard and state-of-the-art variants of MIL. First, slide images were sliced into nonoverlapping 224 × 224 patches as instances for the following training procedure. In the standard MIL approach, we evaluated the performances when $k$ was set as 1, 3, or 5, where $k$ was the number of patches selected per slide during the training phase. Variants of MIL included expectation maximization-based methods[19] with logistic regression and support vector machine slide-level aggregation, denoted by EM-CNN-LR and EM-CNN-SVM, respectively. Furthermore, MIL with max feature aggregation and random forest slide-level aggregation, denoted by CNN-MaxFeat-based RF[24], and MIL with recurrent neural network slide-level aggregation, denoted by MIL-RNN[23], were evaluated.

The model performance results are listed in Table 1. Among standard MIL models with different $k$ values, the MIL model with $k = 3$ achieved the optimal testing AUC scores of 0.9188 (0.9052–0.9324) and 0.9032 (0.8825–0.9240) for adenocarcinoma and squamous cell carcinoma classification, respectively. This result indicated that $k = 3$ is an appropriate bag size that can ensure that true positive patches are sampled in positive cases while patches that might be normal tissue are not sampled in cases of a small cancerous lesion.

By contrast, EM-CNN-LR and EM-CNN-SVM achieved AUC scores of 0.76–0.83, meaning they performed worse than standard MIL. This suggests that the interpatch aggregation method, which sums class probabilities of patches, may not be suitable for a heterogenous data set—namely, one composed of both resections and biopsies. Summing patch results tends to result in slides with numerous positive patches being classified as positive and those with few positive patches being classified as negative. Our data set, however, consisted of biopsy and resection slides, and the lesion sizes in the biopsy slides were much smaller and likely to be ignored.

Finally, both CNN-MaxFeat-based RF (AUC = 0.9345, $P = 2.938e-4$ for adenocarcinoma and AUC = 0.9071, $P = 0.6459$ for squamous cell carcinoma) and MIL-RNN (AUC = 0.9310, $P = 1.003e-3$ for adenocarcinoma and AUC = 0.9239, $P = 1.472e-3$ for squamous cell carcinoma) outperformed standard MIL ($k = 3$), demonstrating the effectiveness of these patch aggregation methods.

**Model performance of the whole-slide training method**. By leveraging the UM mechanism, our proposed whole-slide training method trained and evaluated an entire 21,500 × 21,500 image end-to-end without any patching procedure and additional aggregation models to derive slide predictions. Moreover, the ResNet-50 with two variants of final pooling operations, namely global average pooling (GAP) and global max pooling (GMP), were evaluated.

As presented in Table 1, the whole-slide training method with GAP layers achieved AUC scores of 0.6506 (0.6213–0.6798) for adenocarcinoma and 0.5597 (0.5176–0.6018) for squamous cell carcinoma classification, indicating the model only captures limited information from inputs. Although GAP layers are widely adopted in state-of-the-art CNN models[3–6] in most natural image classification methods, their application to ultrahigh-resolution images is prone to losing subtle information presented by tiny features. Such inefficiency leads to significant degradation of

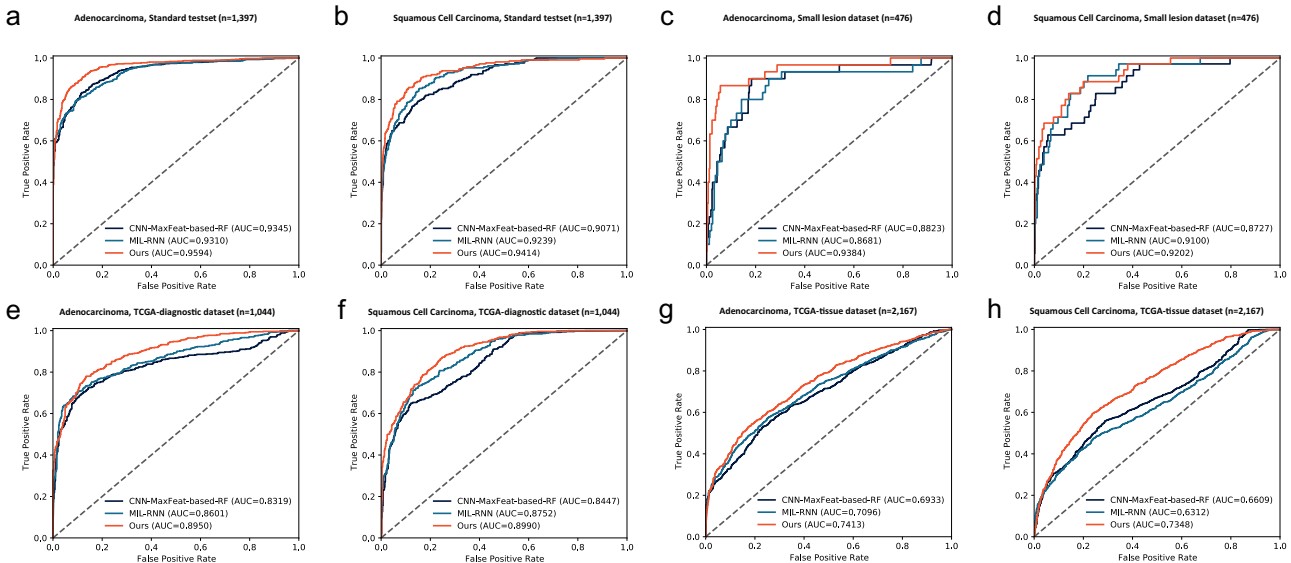

**Fig. 2 Receiver operating characteristic (ROC) curves of methods on classifying adenocarcinoma and squamous cell carcinoma.** Area under the ROC curve (AUC) was measured for each data set and method. All the measurements were taken from distinct samples. **a**, **b** ROC curves on the standard testing data set ($n = 1397$). **c**, **d** ROC curves on the small lesion data set ($n = 476$). **e**, **f** ROC curves on TCGA-diagnostic data set ($n = 1044$). **g**, **h** ROC curves on TCGA-tissue data set ($n = 2167$). The dark blue, cyan, and red lines represent CNN-MaxFeat-based RF, MIL-RNN, and whole-slide training method, respectively.

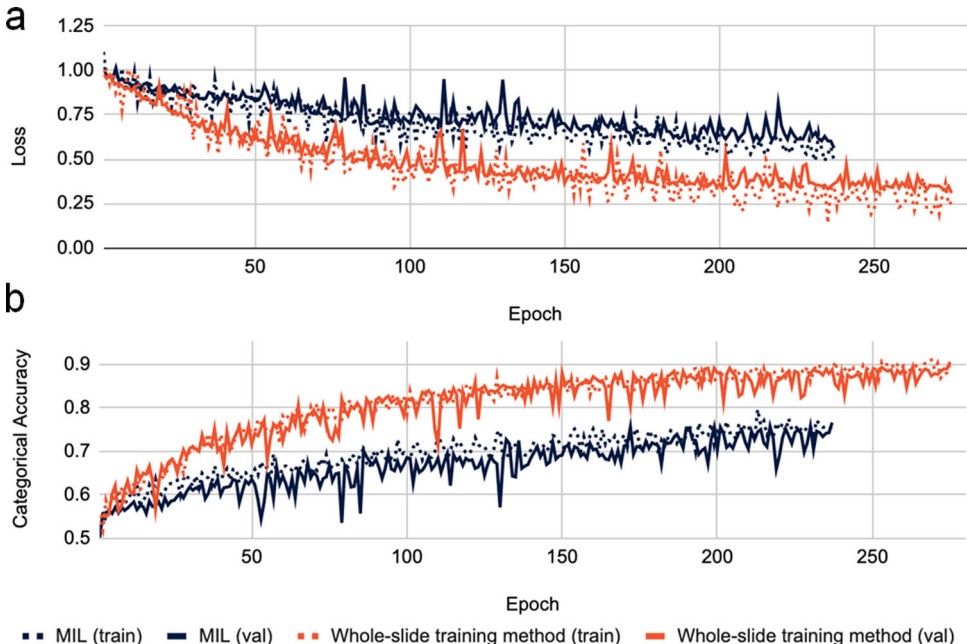

**Fig. 3 Learning curves of MIL and whole-slide training method.** The lines represent the loss (**a**) and the accuracy (**b**) of MIL (dark blue) and whole-slide training method (red) for the training set (dotted) and the validation set (solid). The x axis represents the numbers of elapsed training epochs.

model performance compared with the whole-slide training method with GMP layers.

By contrast, the whole-slide training method with GMP layers achieved AUC scores of 0.9594 (0.9500–0.9689) for adenocarcinoma and 0.9414 (0.9234–0.9593) for squamous cell carcinoma, which were significantly superior to those achieved by the other approaches, including CNN-MaxFeat-based RF ($P = 3.565e{-}7$ and $3.189e{-}4$ for classifying adenocarcinoma and squamous cell carcinoma, respectively) and MIL-RNN ($P = 2.279e{-}9$ and 0.02498 for classifying adenocarcinoma and squamous cell

carcinoma, respectively). Receiver operating characteristic (ROC) curves for the aforementioned models are presented in Fig. 2a, b.

As illustrated in Fig. 3, the learning curves of the MIL method and our proposed method demonstrated significantly different patterns in the early training stage. The performance of our proposed method increased sharply during the first few epochs and converged gradually in the remaining training time. This is a typical learning pattern when training DNNs because models tend to seek and learn features that can be easily divided into high

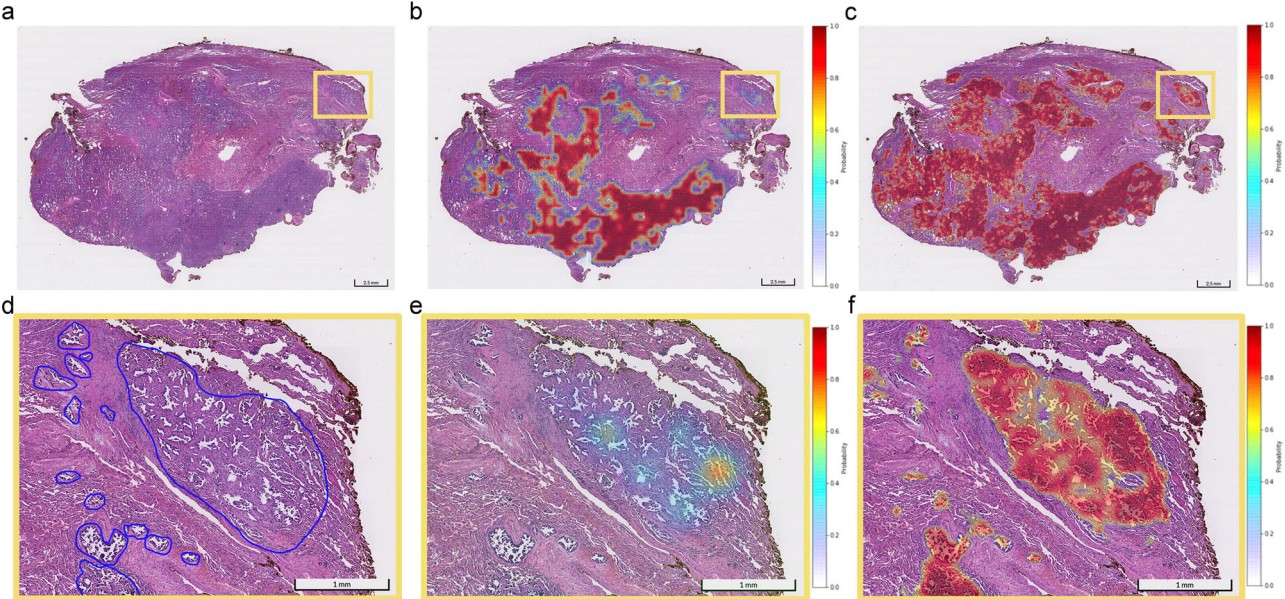

**Fig. 4 Visualization of heatmaps generated by models.** Both heatmaps are upscaled to fit the original image size using a bicubic interpolation. The colors of the overlaid heatmaps represent the predicted probabilities of being a tumor, as defined in the color bar. **a** Whole-slide view on a slide containing adenocarcinoma lesions. **b** Heatmap of MIL on a whole-slide view. **c** Heatmap of whole-image method. **d** Zoom-in view with human annotations (blue contours). **e** Zoom-in view of MIL heatmap. **f** Zoom-in view of heatmap generated by whole-slide training method.

dimensional representation spaces first, which contributes to a drastic improvement in accuracy. Along with the training procedure, most of the obvious features were used and model performance saturated gradually. Subtle features were extracted in later training stages because the models attempted to minimize losses through seeking high dimensional planes, which led to further refinement of the models. By contrast, the learning curves of MIL were relatively smooth during the first few epochs. The MIL training procedure relies on its work-in-progress model to choose representative tiles from WSIs before training the classifier. However, models in the early training stage exhibited random-guess behavior because they had not yet learned any information. Those wrongly selected tiles would inevitably misguide the models and thus slow the convergence rate.

**Visualization**. Although CNNs have achieved impressive performance in classification tasks, more intriguing is how the models make decisions. Visualization is the most straightforward approach for investigating how models learn to solve a given task. Different visualization approaches were applied to different models because the internal properties were not the same between the MIL and whole-slide models.

For the MIL model, prediction maps of slides could be simply derived through assembling the probabilities of tiles forwarded by the patch-level classifier. For the whole-slide training method, the CAM[30] technique was adopted to visualize discriminative regions related to particular categories of cancer.

As depicted in Fig. 4, both the MIL model and whole-slide model could discover representative information, which was highlighted by heatmaps after iteratively learning from slide-level diagnosis. Furthermore, our method, coupled with CAM, revealed a more comprehensive ability to highlight all suspicious areas on the slide, especially small lesions.

**Model performance on the small lesion testing data set**. To investigate the performances of different models on hard cases, we picked 69 slides with small lesions (i.e., tumor area < 10% of tissue area) along with 407 non-cancer slides from the standard

testing data set. Slides in this sub-data set were particularly prone to misclassification by pathologists because of the small percentages of cancerous tissue regions. We compared the whole-slide training method, CNN-MaxFeat-based RF, and MIL-RNN on this subset.

The ROC curves and AUC scores of the three models in Fig. 2c, d and Table 1 indicate the performances on the small lesion testing data set. The results demonstrated that slides with a small lesion were more difficult to distinguish; moreover, the AUC scores of all models dropped to some extent—from 0.01 to 0.07—when compared with those tested on the standard testing data set. Nevertheless, our method exhibited superior performance on slides with small lesions when compared with CNN-MaxFeat-based RF ($P = 0.1171$ for adenocarcinoma and $P = 0.08036$ for squamous cell carcinoma) and MIL-RNN ($P = 0.02924$ for adenocarcinoma and $P = 0.5760$ for squamous cell carcinoma).

**Model performance on Cancer Genome Atlas data sets**. To examine the generalization ability of the models, two lung cancer data sets, namely LUAD (lung adenocarcinoma) and LUSC (lung squamous cell carcinoma) from The Cancer Genome Atlas (TCGA) public data set, were included in the study. The TCGA-LUAD and TCGA-LUSC diagnostic slide data consist of 532 slides of adenocarcinoma and 512 slides of squamous cell carcinoma, respectively. Because color characteristics varied greatly between slides in the TCGA diagnostic data set and our data set, we applied the Vahadane stain normalization algorithm[32] as part of image preprocessing to match the color spectrum distribution of the training slides with that of the TCGA slides[32–34]. We compared our method (with GMP layers), CNN-MaxFeat-based RF, and MIL-RNN on this subdataset, all of which were trained on the whole training data set.

As illustrated in Fig. 2e, f and Table 1, AUC scores were reduced when testing on the TCGA diagnostic data set for all methods. Specifically, our method achieved an AUC score of 0.8950 (0.8764–0.9137) for classifying adenocarcinoma on TCGA diagnostic slides, which is worse than the score on the standard

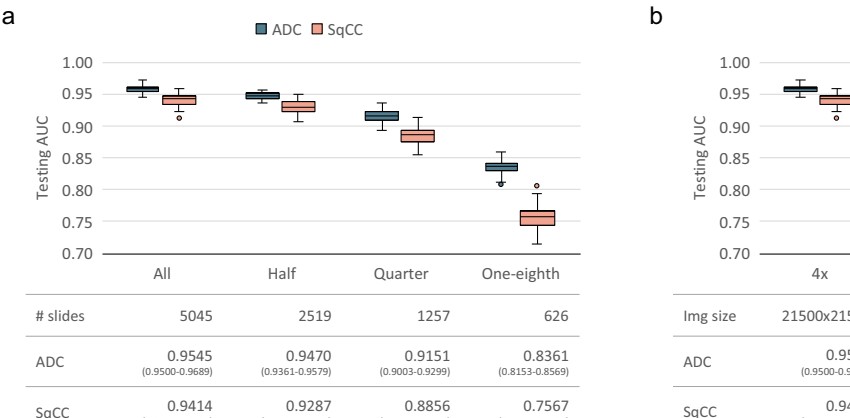
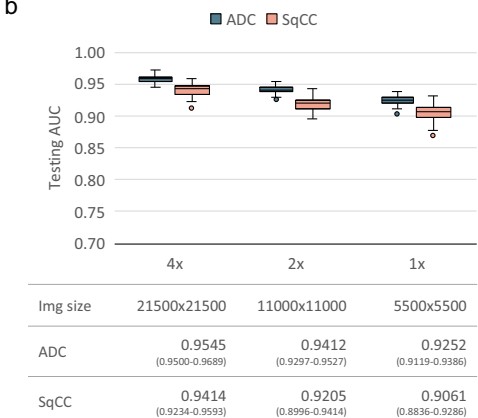

| | All | Half | Quarter | One-eighth |
|---|---|---|---|---|
| # slides | 5045 | 2519 | 1257 | 626 |
| ADC | 0.9545 (0.9500-0.9689) | 0.9470 (0.9361-0.9579) | 0.9151 (0.9003-0.9299) | 0.8361 (0.8153-0.8569) |
| SqCC | 0.9414 (0.9234-0.9593) | 0.9287 (0.9091-0.9483) | 0.8856 (0.8608-0.9104) | 0.7567 (0.7234-0.7900) |

| | 4x | 2x | 1x |
|---|---|---|---|
| Img size | 21500x21500 | 11000x11000 | 5500x5500 |
| ADC | 0.9545 (0.9500-0.9689) | 0.9412 (0.9297-0.9527) | 0.9252 (0.9119-0.9386) |
| SqCC | 0.9414 (0.9234-0.9593) | 0.9205 (0.8996-0.9414) | 0.9061 (0.8836-0.9286) |

**Fig. 5 Data set size and image resolution impact on whole-slide training method.** The performances were measured by area under the ROC curve (AUC) of classifying adenocarcinoma (ADC, dark blue box-and-whiskers) and squamous cell carcinoma (SqCC, red box-and-whiskers) over the standard testing data set ($n = 1397$). The AUC distribution of each configuration was obtained based on the sampled AUCs by bootstrapping over 100 iterations. A box-and-whisker is drawn to represent each distribution with center (Q2, the median AUC), bounds of box (Q1 and Q3, the first and third quartiles of AUC), bounds of whiskers (the minimal and maximal AUCs within the range obtained by adding Q2 by ±1.5 times the distance between Q3 and Q1), and outlier points (AUCs out the range). The tables below report the exact AUC without bootstrapping and the confidence interval calculated by Delong's method of each configuration. **a** Performance of models trained with reduced numbers of slides. **b** Performance of models trained with lower image resolution.

testing data set (AUC = 0.9594 [0.9500–0.9689], $P = 1.987e−9$). However, the whole-slide training method still demonstrated a greater model generalization ability compared with CNN-MaxFeat-based RF ($P = 2.138e−9$ for adenocarcinoma and $P = 2.849e−8$ for squamous cell carcinoma) and MIL-RNN ($P = 1.033e−4$ for adenocarcinoma and $P = 5.221e−3$ for squamous cell carcinoma).

We further examined model performance on the TCGA tissue data set, which is composed of 1067 and 1100 fresh-frozen section slides of adenocarcinoma and squamous cell carcinoma, respectively. As depicted in Fig. 2g, h and Table 1, AUC scores were significantly lower when the models inferred from the TCGA tissue data set, as was expected. One of the most critical factors is ice crystal artifacts caused by the procedure of freezing sections. These artifacts heavily distort morphological features that are critical for classifying the types of lung cancer. Moreover, given a large domain discrepancy between the distributions of training (formalin-fixed paraffin-embedded [FFPE]-slide domain) and testing data sets (frozen-slide domain), model performance was reduced considerably.

**Impact of data set size and image resolution.** The prediction performance and robustness of DNNs can be improved simply by feeding them more data. However, this also increases the requirement of computational resources. Histopathological images scanned at a ×40 magnification typically have a total number of pixels in the billions. Training DNNs on histopathological images with such extreme resolutions requires staggering amounts of computational resources to iterate through all data. In this study, for example, the number of pixels in the 9662 WSIs was at least 1000 times greater than that of ImageNet images. Campanella et al.[23] went further and collected 44,732 WSIs to train MIL models. Striking a balance between model accuracy and resource conservation is critical when training models on WSIs. We evaluated the trade-off between model performance and data set scale by downscaling the training set size or image resolution of the whole data set. To control confounding factors, the model architecture setup (ResNet-50 GMP with fixup initialization) and hyperparameters were identical across all experiments. Finally, model performance was evaluated using AUC scores on the standard testing data set.

As illustrated in Fig. 5, the reductions in training data set size and image magnification level reduced the models' performance on the testing set. For instance, the testing AUCs of the training model on the full data set (5045 slides) were significantly higher ($P = 7.251e−6$ for adenocarcinoma and $P = 1.222e−2$ for squamous cell carcinoma) than those on half the data set (2519 slides). Similarly, reducing the image resolution to ×2 lowered the AUCs ($P = 1.779e−6$ for adenocarcinoma and $P = 3.249e−3$ for squamous cell carcinoma). These experiment results suggested the need for a massive amount of high-resolution slide data to allow models to capture detailed information, thus providing improved performance.

Notably, even though the reduction in resolution affected model performance, the AUCs of the whole-slide training method on downsampled slides were still high, and those using ×1 magnification were higher than 0.9 (0.9252 for adenocarcinoma and 0.9061 for squamous cell carcinoma). Compared with MIL methods trained using ×4 magnification, our method trained using ×2 magnification achieved a competitive result with CNN-MaxFeat-based RF (AUC = 0.9412 vs 0.9345, $P = 0.2378$ for adenocarcinoma and AUC = 0.9205 vs 0.9071, $P = 0.1994$ for squamous cell carcinoma). The results also indicated that our method trained using ×2 magnification was not significantly inferior to the MIL-RNN method trained using ×4 magnification on the identification of both adenocarcinoma (AUC = 0.9412 vs 0.9310, $P = 0.06015$) and squamous cell carcinoma (AUC = 0.9205 vs 0.9239, $P = 0.6865$). Therefore, given limited computational resources, reducing the resolution is a viable strategy for shortening the training time and acquiring acceptable model accuracy for tasks such as the classification of lung cancer types.

**3-Class classifier versus multiple binary classifiers.** According to information theory, the output probabilities of most classification algorithms are designed to minimize the entropy regardless of the number of target classes, meaning that splitting a single multiple-class classification task into multiple binary classification problems may not lead to better results. In the current study, splitting a single model that identifies patterns of adenocarcinoma, squamous cell carcinoma, and non-cancer into multiple individual models did not achieve better results. Our experiment results revealed no significant differences between individual classifiers

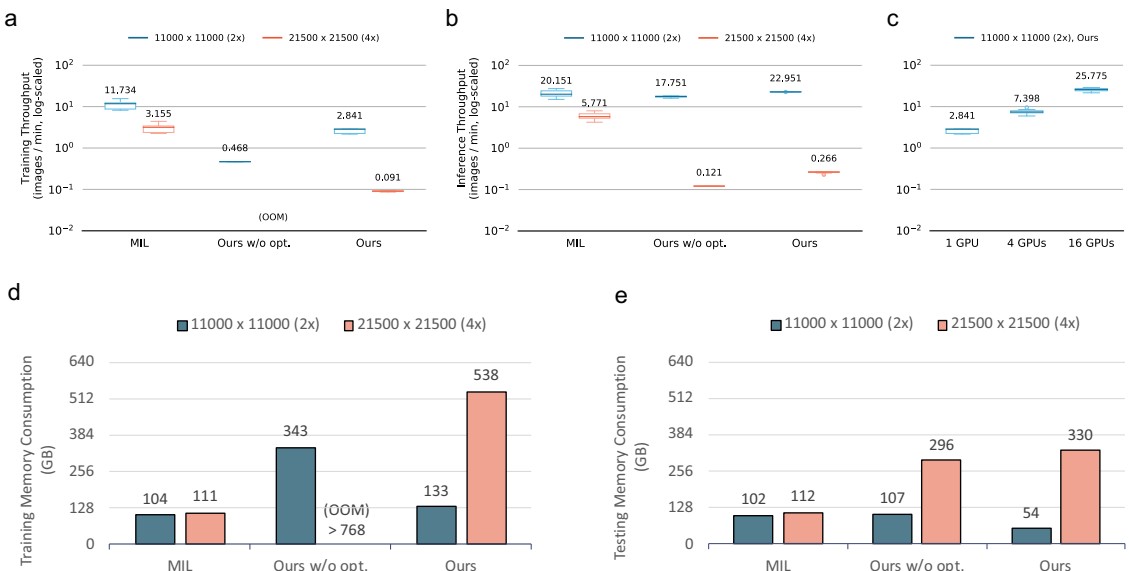

**Fig. 6 Throughput and memory consumption of training and inference on our proposed method against non-optimized version and MIL.** For throughput tests, we measured the execution time of 30 training iterations of each configuration to acquire the distribution. All boxes-and-whiskers in **a**–**c** represent a distribution with center (Q2, the median), the bounds of box (Q1 and Q3, the first and third quartile), and the bounds of whisker (the minimal and maximal values within the range obtained by Q2 ± 1.5 × (Q3 – Q1)). **a** Training throughputs of whole-slide method, non-optimized version, and MIL on 11,000 × 11,000 (blue) and 21,500 × 21,500 (red) images with a single GPU. Note that the configuration of training 21,500 × 21,500 images by non-optimized whole-slide training method is not executable due to incurring out-of-host-memory error. **b** Inference throughputs of models with a single GPU. **c** Training throughputs of our method with multiple GPUs. **d** Host-memory consumption of our method w/ and w/o optimization and MIL on 11,000 × 11,000 (dark blue) and 21,500 × 21,500 (red) images during training phase. Training 21,500 × 21,500 images by non-optimized whole-slide training method encounters an out-of-memory error, implies the memory consumption is higher than the host-memory capacity of a node, 768 GB. **e** Host-memory consumption of models during inference phase.

and the 3-class model. The AUC scores were 0.9548 (0.9450–0.9646) for adenocarcinoma and 0.9414 (0.9239–0.9588) for squamous cell carcinoma.

**Throughput comparison and memory consumption.** Subsequently, we compared the computing throughputs of training and inference among the different methods by measuring the number of slides processed per minute. All experiments were conducted on two image resolutions, namely 21,500 × 21,500 (×4 magnification) and 11,000 × 11,000 (×2 magnification). Specifically, we adopted standard MIL with $k = 1$ as the representative to measure performance because throughputs among variants of MIL methods are close. Our proposed method fit images directly without dividing the input. To avoid the out-of-memory problem, we leveraged and optimized the UM mechanism to offload temporary data to host memory efficiently through graph editing and mixed precision[35]. Compared with vanilla UM, the throughput of the whole-slide training method could be accelerated by 6.26× through the incorporation of both optimized memory access and mixed-precision training, as illustrated in Fig. 6a. Moreover, our proposed method can integrate orthogonally with synchronized data parallelism, the most commonly used distributed training approach, for further acceleration. We conducted experiments on TAIWANIA 2 with a hardware configuration of 16 GPUs; the training process achieved a 64.60× throughput compared with a single GPU, non-optimized one, as illustrated in Fig. 6c.

The MIL method follows the patch-based protocol and patches can be placed on GPUs without additional CPU–GPU swapping, and high efficiency is retained during both the training and inference phases. As illustrated in Fig. 6, throughputs of the MIL method were 3.5× and 15.7× faster

than our proposed method under the training phase when the sizes of inputs were 11,000 × 11,000 and 21,500 × 21,500, respectively. Notably, the throughput of our method was 1.3× faster than that of the MIL method when the size of the input was 11,000 × 11,000, whereas it was 16.4× slower when the size was 21,500 × 21,500 during the inference phase. This suggests that the memory overhead of our method did not expand linearly, resulting in a throughput gap between different input sizes. On average, our method took 100 GPU-days and 1200 GPU-days to reach convergence when experimenting with ×2 and ×4 magnification, whereas the MIL method took 30 GPU-days and 120 GPU-days in the training phase. As for the inference phase, our proposed method took 1.5 GPU-hours and 80 GPU-hours to complete 1397 WSIs with ×2 and ×4 magnification, whereas the MIL method took 1.5 GPU-hours and 5 GPU-hours for the same experimental conditions.

Although our proposed method is not constrained by GPU memory size because UM is enabled, adequate host-memory space is required to store intermediate data produced during model training. We measured the host-memory consumption among the different methods on both 11,000 × 11,000 and 21,500 × 21,500 inputs by subtracting the peak host-memory usage from that during an idle state. As indicated in Fig. 6, the memory consumption of the MIL method remained constant in the training and inference stages on both image sizes. No matter how large an input image is, MIL processes one patch at a time, requiring a fixed size of temporary memory space for patch processing. By contrast, the memory consumption of our method scales with the input resolution to store global context information. With memory optimization and mixed-precision training, our method consumed less than half of the memory consumed by the non-optimized version during training.

## Discussion

Patch-based methods have been popular in deep learning for digital pathology because they readily circumvent the memory constraints of compute accelerators. Supervised patch-based methods require patch-wise annotations that are not plausible to employ in routine pathology. Moreover, generating detailed annotations on WSIs is extremely laborious; an expert can take 1 h to annotate just portions of a single WSI. Furthermore, borders between tissue types are often ambiguous, leading to inconsistencies between pathologists. The high variability of tissue morphology makes it difficult to cover all possible examples during annotation. These shortcomings expose deep learning models to strong bias from expert-defined annotations and make them difficult to learn comprehensively. To avoid the annotation burden and selection bias of experts, most recent studies have applied weak supervision methods, which can train deep learning models to explore relationships inside WSIs, resulting in direct clinical diagnoses.

Training cancer classifiers without detailed annotations alleviates the burden of experts and allows DNN models to benefit from numerous WSIs with readily available sign-out diagnoses. In previous studies, methods for detecting cancer using strongly supervised models trained with patch-wise annotations[11–19] still outperformed weakly supervised models. However, research on weakly supervised models for cancer detection is gaining popularity because annotation is too costly and because models trained through strong supervision are limited by how targets are annotated.

Although MIL can be trained with weak labels such as slide diagnosis, several drawbacks may possibly affect model performance: (1) Incorrectly selected patches in early training iterations may trap models in the local minimum because of random initialization. In some situations, models even stop improving after the first few epochs. (2) MIL attempts to utilize the $k$-most representative patches while abandoning other related information contained in others. However, the size of the $k$-most representative patches is a hyperparameter and the truly informative regions may vary from slide to slide. The $k$-most representative patches may either overtake irrelevant patches as positives or lack the capacity to include atypical patterns that are crucial for diagnoses.

Instead of modifying algorithms and the training pipeline toward weak supervision, thus avoiding the out-of-memory problem, we consider it more straightforward to leverage the UM mechanism to train CNNs directly with numerous images without modifying the training pipeline. UM enables GPUs to access the host memory directly, which expands the capacity from gigabytes to terabytes. The UM mechanism works by swapping data between GPU and host memory, which is slow because of frequent data exchange through the Peripheral Component Interconnect Express (PCIe) interface. Limiting the use of working memory can result in significant acceleration, as we demonstrated in this study. However, training CNNs on images larger than 20,000 × 20,000 pixels is prohibitively slow. Eventually, limitations in host memory will occur. Therefore, further research into more memory-efficient algorithms or training methods is necessary. The present research processed WSIs at a magnification equivalent to a ×4 objective lens. To address image recognition problems that require resolutions at ×20 or ×40 magnification, it is likely beneficial to take a two-step approach: first, a ×4 image should be used to locate crucial regions in the WSIs, and second, ×40 images of those regions should be used for the final image recognition task. This process would closely resemble the real-life practice of pathologists.

Campanella et al.[23] demonstrated that MIL can achieve excellent results by using a WSI data set with only slide-level labels. In this study, we demonstrated that using entire WSIs for training can achieve superior results. Two possible explanations may account for this. The first is that given the randomness in the sampling process, MIL requires a much greater number of training samples to reach the same level of performance. The second is that, given the lack of ground truth at the patch level, a ceiling exists to the performance MIL can achieve. Therefore, further research comparing the two methods using data sets with tens of thousands of WSIs may be necessary.

The lesion localization of our model through CAM revealed good coverage in most cases. Notably, however, the semantics of CAM in our proposed method was slightly different from cancer cell location. Areas highlighted by CAM were highly related to predictions; DNNs use distinguishable features across the given data set to classify an image into groups, which may introduce frequent side effects such as contextual bias. For instance, a classifier trained to learn cars and boats will highlight not only the boat itself but also water because boats are always accompanied by water. In our case, the CAM for squamous cell carcinoma highlighted not only cancerous regions but also necrotic regions (Supplementary Fig. 1). Despite the fact that necrosis can be caused by diseases other than squamous cell carcinoma, such as physical injury or infection, these diseases rarely mandate biopsies and therefore were underrepresented in the training data set. The model thus learned necrosis to be an identifying feature of squamous cell carcinoma.

Because deep learning models collect all possible clues to make decisions, the model unavoidably learns that such weak relations, or halo effects, are useful when differentiating adenocarcinoma from squamous cell carcinoma. Similar to Li et al.[36], who added extra supervision to guide attention maps, one method for resolving this problem is to add a small number of slides annotated in detail to specify cancer cells. These annotations provide hints for models to separate cancerous representations from weakly related representations. In future studies, such integration for leveraging both slide-level annotations and limited-detail annotations can be developed to achieve a more comprehensive and precise model.

Compared with the development of artificial intelligence (AI) in radiology, progress in pathology AI has been slow. We reason that this is not for the lack of data but rather the burden of annotation. A large medical center can generate up to half a million slides per year, yet a typical digital pathology AI project uses only hundreds of slides. The scope of a pathology AI project is most often constrained by annotation, which is inherently slow because it is performed at a microscopic level. Our methods pave the way for more rapid progress in pathology AI research through reducing the need for detailed annotation. Our method can be applied to various classification tasks and potentially even to multilabel learning—for example, for determining the presence of multiple tissue subtypes in a lung tumor section. We expect our method to be most useful when it can be combined with strongly supervised methods, which would enable it to leverage a large amount of weakly labeled data to achieve a crude understanding; subsequently, a small strategically annotated data set could be used to fine-tune its performance, thus achieving superior lesion localization and precise semantics in multilabel classification or subclassification tasks.

## Methods

**Data set**. A total of 9662 hematoxylin and eosin (H&E) stained formalin-fixed paraffin-embedded (FFPE) specimens collected from 2843 patients were retrieved from TMUH, WFH, and SHH within the period from 2018 to 2019. For each case, it might have at least one H&E slide gathering from either biopsy or different parts of the resected lung tissue and diagnosed as the dominant type of cancer or non-cancer tissue. Lung specimens were sampled by either biopsy (32%, 3075 slides) or resection (68%, 6587 slides).

### Table 2 Case numbers of digital slides collected from TMUH, WFH, and SHH.

|  | Non-cancer | Adenocarcinoma | Squamous cell carcinoma |
|---|---|---|---|
| Training set (total: 5045 slides) | | | |
| TMUH | 953 | 401 | 98 |
| WFH | 509 | 1123 | 318 |
| SHH | 7 | 1271 | 365 |
| Validation set (total: 561 slides) | | | |
| TMUH | 103 | 46 | 11 |
| WFH | 58 | 123 | 38 |
| SHH | 2 | 138 | 42 |
| Testing set (total: 1397 slides) | | | |
| TMUH | 264 | 111 | 27 |
| WFH | 141 | 311 | 88 |
| SHH | 2 | 352 | 101 |

A total of 7003 slides were collected in the current study, including 2039 cases of non-cancer, 3876 cases of adenocarcinoma, and 1088 cases of squamous cell carcinoma.

Because of a lack of samples in certain types of rare lung cancer, we filtered out cases that have less than 500 samples. As a result, the cases diagnosed as small-cell carcinoma and large-cell carcinoma were excluded. The final data set contains 7003 slides, including 3876 cases of adenocarcinoma, 1088 cases of squamous cell carcinoma, and 2039 cases of non-cancer tissues. The diagnoses were confirmed by at least two pathologists.

The data set was randomly split into training, validation, and testing sets, containing 5045, 561, and 1397 slides, respectively, using a stratified sampling method. Detailed numbers of slides from each site are listed in Table 2. Unless otherwise specified, experiments were conducted using this cross-site data set configuration.

To investigate the model performance on hard cases, we used only cancerous cases that have small lesions (i.e., tumor area < 10% of tissue area), in addition to benign cases, of the testing set. This subset comprises 476 slides in total, including 30 slides of adenocarcinoma and 39 slides of squamous cell carcinoma, and 407 slides of benign cases.

The access of data was compliant with policies (NO.1080506_1) approved by the Office of Legal Affairs, Taipei Medical University and National Legislation For Research. The data collections were allowed proceeding after informed consents were obtained from participants. All WSIs were de-identified and do not contain any patient information or label text except for slide-level diagnosis. The collected data are limited to research use only.

All slides were scanned by Hamamatsu NanoZoomer XR in ×20 magnification (0.46 μm per pixel). The original resolution of each slide image differs with width up to 110,592 pixels (average: 65,683 pixels) and height up to 55,552 pixels (average: 40,593 pixels).

To evaluate cross-site generalization ability of models, the TCGA lung cancer slides were also included in the current study. The TCGA-diagnostic data set contains 532 H&E stained FFPE tissue slides of adenocarcinoma from TCGA-LUAD diagnostic data (9 slides were abandoned since no magnification information was provided) and 512 slides of squamous cell carcinoma from TCGA-LUSC diagnostic data. All the slides were scanned by Aperio scanner in either ×20 (0.50 μm per pixel) or ×40 magnification (0.25 μm per pixel). For resolution consistency, these slides were rescaled in proper factor to meet the pixel spacing with that of our slides (0.46 μm per pixel). Furthermore, we used the stain normalization method proposed by Vahadane et al.[32] to unify the color style of the TCGA slide images. For the supplementary study of model performances on non-FFPE slides, frozen sectioned slides from TCGA-LUAD and TCGA-LUSC tissue data sets were evaluated, including 1,067 slides of adenocarcinoma and 1100 slides of squamous cell carcinoma cases.

**Multiple-instance learning**. Most image binary classification tasks can be formulated into a MIL problem by dividing an image into multiple partial regions if the following criterion is met: a bag, or an image, is labeled as positive when the target shows up in at least one instance, or local region; and labeled as negative if the target is absent from all instances. Hence, any positive bag can be represented by using several critical instances only whereas no suspicious instances should be left in negative bags.

This property makes training models with bag-level labels become possible by applying positive bag-level labels to critical instances in positive bags and negative bag-level labels to all instances in negative bags. During training, the MIL iteratively selects high-score instances from each bag when training a classifier.

To be more specific, the MIL method can be separated into two alternative steps: instance selection and classifier optimization, as illustrated in Fig. 7. During instance selection, an instance selector, which computes the probability of positive

over instances of bags, is used to mine k-most positive instances from each bag. With selected instances, a classifier is trained to maximize the probability of instances selected from positive bags while minimizing the probability of instances selected from negative instances.

In the end of MIL, the classifier will be able to mine the most relative patterns of positive cases and the label of any given bag can be inferred by naive aggregation methods such as taking the maximum scores (max pooling) or averaging scores of $k$-most instances.

Deciding numbers of instances to represent a given image is an issue without consensus. Typically, aggregating with max pooling (i.e., $k = 1$) is the most common implementation among related literatures, especially during the training phase[23], since a positive bag-level label only implies at least one positive instance. Setting $k > 1$ may lead to instances oversampling when true positive instances are less than $k$ in a given bag, making the model tend to predict as negative on these bags. Though the side-effect of using $k > 1$ may occur in some situations especially when the critical pattern is relatively tiny, setting an appropriate size of $k$ greater than 1 is shown to be more robust since the model is less likely to be affected by outlier instances. To strike the balance, many variants of MIL were proposed. Expectation-maximization (EM) MIL[19] is proposed to dynamically select $k$ by leveraging spatial relationships among instances during model training and evaluation. Several works[19,23,24] train a bag-level aggregation model to combine the prediction results of all instances.

**MIL methods on lung cancer type classification**. The lung cancer classification task with only slide-level labels can be considered as a MIL problem since a slide is labeled as either adenocarcinoma, squamous cell carcinoma, or non-cancer tissues.

As illustrated in Fig. 7, we treated each slide as independent bags and cropped patches of 224 × 224 pixels inside each bag as instances to train a classifier and aggregated prediction of instances using the widely adopted max-pooling method. For the instance classifier, a ResNet-50[3] with fixup initialization[31] was implemented. While the dominant lung cancer type classification mainly relies on inspecting tissue-level morphology rather than cell-level morphology as shown in Fig. 1, we generate instances at ×2 and ×4 magnification to provide sufficient observation scope for the classifier. During model training, we applied the following data augmentation to improve its robustness: flipping, translation, rotation, and color augmentations, including random contrast (multiplication by 0.5–1.5), brightness (multiplication by 0.65–1.35), hue (addition by −32–32) and value (addition by −32–32).

Additionally, instances belonging to the background (i.e., all the RGB values larger than 220) were ignored, which drastically reduced the total number of instances by 80% and speeded up the whole training process. After background removal, 9.2 million, 1.0 million, and 2.6 million tiles were included in the training, validation, and testing data set, respectively. To select representative instances of each bag, we set $k = 1$, 3, or 5 in different trials to pick up instance(s) that is/are most likely to be either adenocarcinoma or squamous cell carcinoma instance and mark instances of bags to its corresponding slide-level annotations.

Several previous works, including the EM-CNN-LR, EM-CNN-SVM, CNN-MaxFeat-based RF, and MIL-RNN, were also implemented as benchmarks for our proposed method. EM-CNN-LR[19] and EM-CNN-SVM[19] both leverage expectation maximization (EM) to dynamically tune $k$. During each training iteration, the patch prediction map is firstly applied to Gaussian blurring with a 3 × 3 rectangle kernel. Representative tiles were selected if their prediction scores on the blurred prediction map were higher than image-level threshold, 0.1th-percentile score among patches in an image, or class-level threshold, 0.05th-percentile score of a class. To train the bag-level model, class histograms of patch prediction results are calculated by summing up all the class probabilities of all the patches, and then are fed into a bag-level classifier, a logistic regressor (LR), or a support vector machine (SVM) with radial basis function (RBF) kernel, for final predictions. Instead of averaging values directly, CNN-MaxFeat-based RF[24] trains an additional random forest aggregation model on top of patch results. Specifically, feature vectors were extracted by selecting patches of highest probability from each block through a patch-level model, which is trained by standard MIL with $k = 3$. These feature vectors are then averaged and fed into a random forest model for final predictions. Finally, the MIL-RNN[23] method trains a recurrent neural network (RNN) with 128 hidden units to aggregate top-$k$ instances selected by a standard patch-level model. To fairly compare the performances with each other, all the MIL variants adopted ResNet-50 with fixup initialization among all the experiments.

**Whole-slide training method**. As a workaround algorithm for hardware memory constraints issue, the MIL alters typical CNNs training pipeline and thus produces several drawbacks. First, instances were nearly randomly selected in the early training phase because of the random initialization of the classifier. These selected instances in the early stage strongly affect the trend of the selection strategy of the classifier in the following training process. In some situations, wrongly selected instances, for instance, accompany hyperplasia tissues but not cancer cells itself, may lead the classifier to fall into a local minimum and thus stop improving after a few epochs.

Second, the top-$k$ representative instances may undertake informative regions or overtake irrelevant regions into account, which limits models to learn targets comprehensively. Since overtaking irrelevant instances as positives could be more

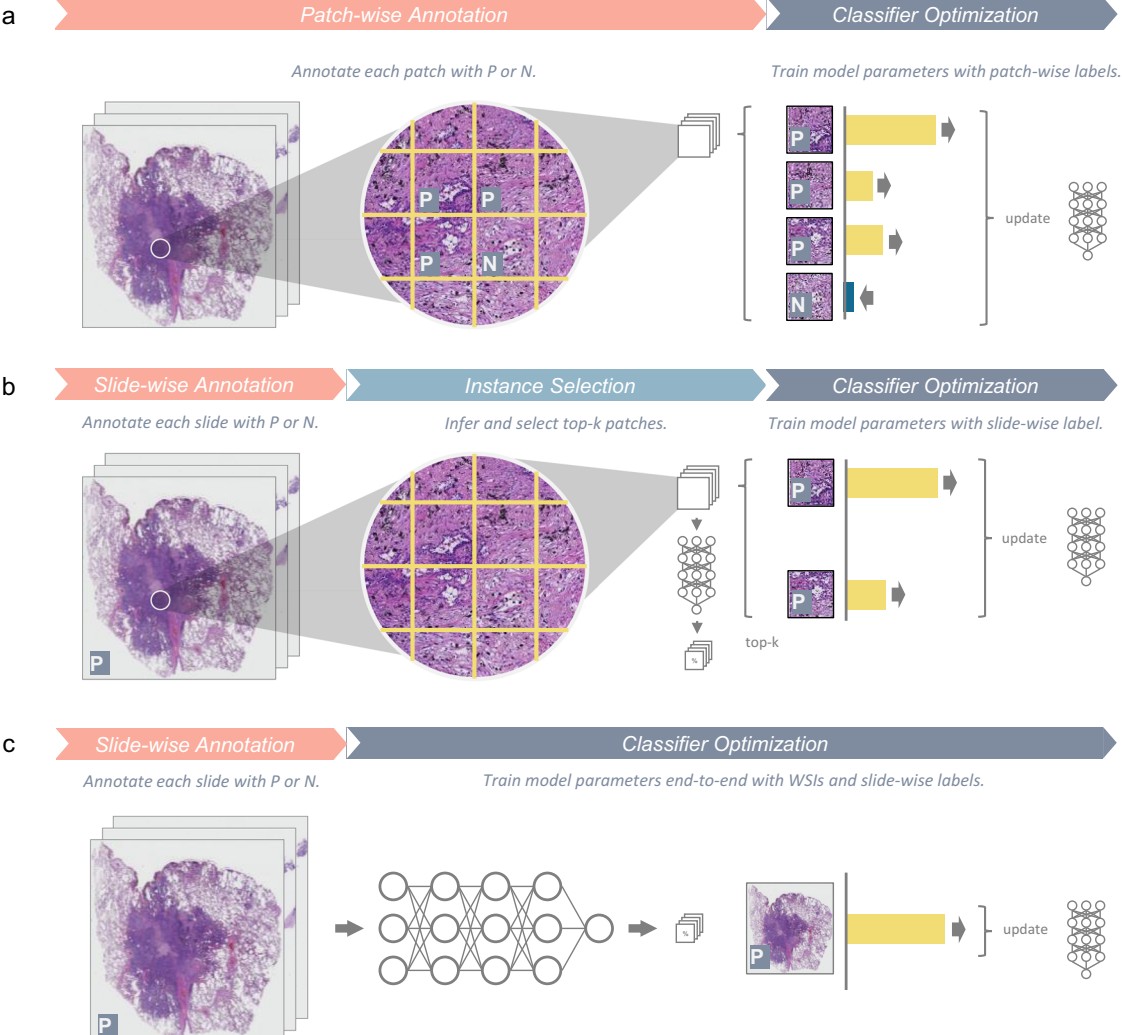

**Fig. 7 Workflow overview of the traditional patch-based method, MIL method, and our proposed whole-slide training method. a** Patch-based method requires a substantial number of patch-level annotations by human experts. Given pairs of a patch image and its corresponding label in manually annotated regions, a deep neural network is trained in a strong supervision manner. **b** MIL leverages slide-level annotations by exhaustive inferences patches and paired the top-k patches most likely to be cancerous to their slide-level tags. These weakly paired patches and temporary tags are then used to update a neural network. **c** Our proposed whole-slide training method feeds an entire slide image with its corresponding slide-level tag into a neural network to fulfill end-to-end slide-level training in a strong supervision manner.

severe to the training procedure, most implementations set $k = 1$ to take the most relevant instance only, which turns out that models will lack capacities to include atypical patterns that are crucial to diagnosis.

By contrast, we propose a whole-slide training method that incorporates the standard CNN architecture with the unified memory (UM) mechanism to support inputs of hundreds of millions of pixels directly to train the models as usual (Fig. 7). Because convolutional layers generate outputs by doing numerous local transformations over height and width dimensions of input, increasing input sizes merely brings more local transformation loadings to CNN models, and therefore, is algorithmically feasible for using whole slide images as inputs. However, the memory consumption scales with the extremely high image resolution of slides will exceed the limit of GPU memory easily. By analyzing memory footprints, it is obvious that not all tensors are required at the same time; therefore, offloading tensors out of GPUs can reduce the in-time resource demands for GPUs.

As a CUDA feature, unified memory (UM) allows GPUs to grant direct access to host memory, which provides terabytes of memory instantly to accommodate most intermediate tensors during forward- and back-propagation. Basically, UM shares the same idea with virtual memory. A unified memory space consists of pages, each of which is virtually addressed and physically stored in either GPU or host memory. Once the total number of pages is out of maximal GPU capacity, a limited number of pages can be placed on the GPU memory, whereas the rest pages will be placed on the host memory. Pages stored in GPU memory can be directly accessed by GPU cores. Otherwise, the system will trigger an on-demand data migration that moves the targeted pages from host to GPU memory beforehand. Oftentimes, when GPU memory is full, a page on GPU memory will be evicted to

host memory simultaneously to spare space for the requested pages. Such a process, called data swapping, allows GPUs to access all content in a unified memory allocation as long as there is enough host-memory space for swapping. Introducing UM in Tensorflow can simply be fulfilled by replacing all GPU memory allocation requests, invoked by cudaMalloc, by unified memory allocation, invoked by cudaMallocManaged. Because UM is transparent, all the tedious swapping operations are done in the background and no other modifications should be made for Tensorflow.

**Performance optimization of whole-slide training method**. While UM circumvents the problem with memory constraint, the frequent data swapping between the host memory and GPUs through a slow hardware link, PCIe, tremendously slow down the training throughput. To address this, two memory optimization techniques, Group Execution and Group Prefetch, were proposed to increase the efficiency by manipulating data swapping during runtime.

Group Execution encourages data swapped in GPU memory to be exhaustively used before swapping out to host memory to reduce the swapping amount. During the training process of most deep learning frameworks, multiple operations are executed in parallel to leverage all computing resources. Because each operation requires a certain amount of memory space to keep intermediate data, excessive parallelism needs more space, sometimes larger than that offered by GPU. At that moment, plenty of data are frequently moved back and forth between GPU and host memory, strongly increasing the swapping amount, referred to as "thrashing". To curb thrashing, Group Execution was implemented to ensure the total memory

consumption of simultaneously executing operations can fit in GPU memory. In Group Execution, neighboring operations are grouped, such that their total intermediate data size is smaller than GPU memory size. Groups are executed sequentially so that concurrent execution of operations only exists within a group scope.

Group Prefetch preloads data required by the upcoming operations in advance to prevent memory stalls. While UM allows GPU programs on-demand access to data stored in host memory when executing an operation, the access is inefficient. First, the transfer rate (3.6 GB/s) of on-demand access is not as fast as that of explicit copy (10.3 GB/s). Second, GPU cores idle when data is being loaded on-demand. To solve the issues, Group Prefetch prefetches the data required by the next group via explicit copy when the current one is being processed. This enables computations and communications to be performed in parallel.

Specifically, the grouping is done before the training process begins. Operations are first organized into a sequence by topological sorting. Second, the grouping program iteratively calculates the total memory consumptions of the first n operations and finds the optimal n just under the memory limit. Third, the first n operations are marked as the first group, and the remaining groups are made by repeating the same process. Finally, the sequential execution between groups is implemented by inserting control dependencies in the operation graph that can be identified by a deep learning framework to force computation ordering. Likewise, prefetch operations are added into the operation graph, each of which uses control dependencies to force the prefetch being called sometime within the duration of a group.

To gain further optimization training speed of our method, we also adopted the mixed-precision training[35] and data parallelism distributed strategy. With mixed-precision training, data except for precision-sensitive values including gradients and feature maps before average pooling layers are stored and computed with 16-bit half-precision floating-point number format, which not only reduces memory requirement but also speeds up computation by using Tensor Cores. In the meanwhile, we used data parallelism distributed strategy to process different samples on multiple GPUs and apply averaged gradients after iterations. To compensate for the loss in randomness between batches caused by the smoothness of averaging gradients collected from different nodes, we set the initial learning rate as the default learning rate multiplied by the square root of the number of GPUs[37].

**MIL assumption in the whole-slide training.** Training a classifier of natural images and a classifier of WSIs are drastically different due to the difference in scale of image size.

Typically, an image of $224 \times 224$ spatial resolution will be condensed into 2048 feature maps of $7 \times 7$ spatial resolution after multiple stacks of convolutional and downsampling layers in the ResNet. Because these layers use sliding window operations, each $7 \times 7$ feature map remains the same spatial arrangement as the input image. To be more precise, these layers can be deemed as a function that each pixel on the feature map encodes a certain size of region on the original input into a single 2048-dimensional embedding vector.

The projection size of a pixel of feature maps corresponding to the original input can be referred to as a receptive field[38]. Information beyond a receptive field has no means to be encoded. According to the operations of ResNet50, the receptive field of the final feature map is $483 \times 483$, which is larger than its common input size: $224 \times 224$. As a result, receptive fields of pixels on the final feature maps have already covered all information of the image.

The following global average pooling (GAP) layer is commonly applied to average feature maps of $7 \times 7 \times 2048$ into a $1 \times 1 \times 2048$ vector. However, the receptive field of any given pixel on the final feature maps will no longer cover the whole image when enlarging the input into tens of thousands of pixels along with its height and width. Such difference is critical in the cancer classification of WSIs. Since malignant regions may be relatively tiny compared to the whole tissues in the positive slides, only very few receptive fields cover critical areas.

With the majority voting aggregation, or the global averaging pooling (GAP), at the end of feature maps, critical signals were further diluted by signals coming from feature maps that are not relevant to patterns of cancers. It ultimately constraints the model to identify slides with small cancerous areas.

Inspired by the MIL, we replace the GAP layer by the global max-pooling (GMP) layer, which only keeps the max value of each element of the 2048-long vectors, as illustrated in Fig. 8. Large values appearing in the embedding vector implies meaningful features are extracted. By adopting GMP, those large values are kept and thus distinguishable signals behind them are preserved.

In addition, having an appropriate size of receptive field is crucial for models to encode necessary information for identifying lung cancer main types. The size of the receptive field of ResNet50 is $483 \times 483$ pixels. Since the receptive field of the model is fixed according to the model architecture, the physical size of receptive fields will vary depending on different magnifications of whole-slide images. To be more precise, the receptive field is around $1111 \times 1111 \ \mu m^2$ and $2222 \times 2222 \ \mu m^2$ for ×4 and ×2 magnification, respectively. While the field of view (FOV) for pathologists to analyze tissue-level morphology at ×100 magnification of a microscope is approximately $200 \times 200 \ \mu m^2$, the receptive fields of our model are sufficient to encode information for identifying lung cancer types.

**Whole-slide training method on lung cancer type classification.** As illustrated in Fig. 7, despite the complexity in the engineering effort for acceleration, the whole-slide training method is logically equivalent to training a deep neural network end-to-end. We used ResNet-50[3] with fixup initialization[31] as the model architecture to keep the same experiment condition as MIL classifiers. WSIs were resized to ×2 and ×4 magnifications and then padded to $11,000 \times 11,000$ pixels and $21,500 \times 21,500$ pixels, respectively, with white color. For each training iteration, each training sample underwent the same augmentation process as that in the MIL training pipeline.

**Experiment setup.** We conducted all experiments on TAIWANIA 2, a multi-GPU, multi-node supercomputing environment. Each node is equipped with 8× Tesla V100 32GB-HBM2 GPUs. The software stack for GPU acceleration included CUDA 10.0 and cuDNN 7.6. We used OpenSlide (version 3.4.1) for slide loading, TensorFlow (version 1.15.3) for model building and training, and Horovod[39] (version 0.19.0), Open MPI (version 4.0.1), and MPI for Python (version 3.0.3) to enable multi-GPU parallel training. All the experiments were executed with batch size 8, 1 sample per GPU. Models are optimized by minimizing 3-class categorical cross entropy as the loss function, calculated by Eq. (1),

$$L = -y_{normal}\ln(\tilde{y}_{normal}) - y_{ADC}\ln(\tilde{y}_{ADC}) - y_{SqCC}\ln\left(\tilde{y}_{SqCC}\right) \quad (1)$$

where $y$ denotes the ground-truth label and $\tilde{y}$ denotes the model prediction on a certain main type.

The weights of all the training models are initialized by ImageNet pretrained weights. Along with the training progress, the kernel weights were gradually updated with the process called gradient descent. We used the Adam optimizer[40] (with an initial learning rate of 2e−5 and decays to 2e−6 when validation loss does not improve in 24 epochs) to train the model and evaluate the performance per 100 training steps as an epoch. During the training process, only the set of weights achieving the lowest validation loss is saved for evaluation.

**Statistics.** We use the area under receiver operating characteristics (Area Under ROC, AUC) as evaluation metrics to measure the slide-level performance of different methods. The 95% confidence interval was obtained using Delong's method[41]. When comparing the AUCs of two models, the P-value was also calculated by the Delong test with a two-sided hypothesis. To evaluate the significance

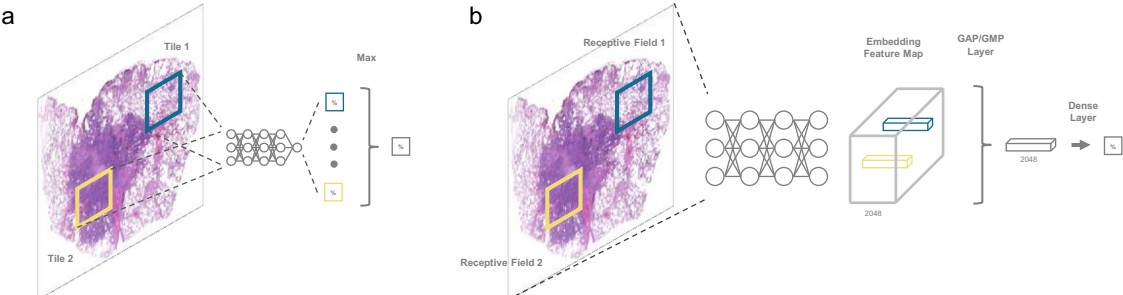

**Fig. 8 Aggregation operations in MIL and in whole-slide training method. a** In multiple-instance learning (MIL), a "max" operation is performed to select the most representative patches by their mapped predicted scores. **b** In the whole-slide training method, the front part of the model encodes the entire image into an embedding feature map, where each vector along the channel axis corresponds to a receptive field. The following GMP (or GAP) layer embeds 2048 max (or average) operations to reduce the spatial dimension. Then, a dense layer performs a linear transformation followed by an activation function on the reduced 2048-length vector for a slide-level prediction.

level of the AUC of a model, we adopted a dummy model that always returns 0.5 as the null hypothesis.

For examining throughputs of different methods, the elapsed times of a model to train on a batch were recorded. We repeated the same procedure 30 times to estimate the distribution of elapsed times.

**Reporting summary**. Further information on research design is available in the Nature Research Reporting Summary linked to this article.

## Data availability

The raw data of models' predictions, learning curves, and throughputs are provided as Supplementary Data 1. The slide data from TMUH, WFH, and SHH are not publicly available due to patient privacy constraints, but are available upon reasonable request from the corresponding author Chao-Yuan Yeh or Cheng-Yu Chen. The slide data supporting the cross-site generalization capability in this study are obtained from TCGA via the Genomic Data Commons Data Portal (https://gdc.cancer.gov).

## Code availability

The source code of this study can be downloaded from https://github.com/aetherAI/whole-slide-cnn [42] and https://github.com/aetherAI/tensorflow-huge-model-support [43] under the CC BY-NC-SA 4.0 license. Whole-slide CNN Training Pipeline provides scripts to reproduce the results in this study, including model training, inference, visualization, and statistics calculation, etc. Also, the pipeline is seamlessly adaptable to other pathological cases. Tensorflow Huge Model Support as a standalone Python library enables high-efficient unified memory training for Tensorflow.

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

## Acknowledgements

This work was supported by grants from the Ministry of Sciences and Technology (grant number MOST108-3011-F-038-001), Taiwan. We thank Dr. Huai-Kuang Tsai and Dr. Trees-Juen Chuang for careful reading and giving advice on this manuscript. We are grateful to the National Center for High-performance Computing for providing computing resources. The results shown here are in part based upon data generated by the TCGA Research Network: https://www.cancer.gov/tcga. This manuscript was edited by Wallace Academic Editing.

## Author contributions

C.-Y.C., C.-L.C, and C.-Y.Y. initiated the study. C.-C.C. and W.-H.Y. designed the experiments and wrote the code. C.-C.C. performed the experiments and analyzed the results. S.-H.C. and C.-Y.Y. reviewed the experiment results. Y.-C.C., T.-I.H., and M.H. critically reviewed and commended the manuscript. All authors contributed to the preparation of the manuscript.

## Competing interests

C.-Y.Y. is the founder, chairman, and chief executive officer of aetherAI. C.-C.C. and W.-H.Y. are data scientists of aetherAI. S.-H.C. is a pathologist of aetherAI. The remaining authors declare no competing interests.
