## [Peer Review File · Nature Communications]

Reviewers' Comments:

Reviewer #1:

Remarks to the Author:

This paper presents a deep learning method for whole-slide pathological classification of lung cancer types by using image-level labels only. Experimental results demonstrate good performance on the collected datasets.

However, I have following suggestions to further improve the manuscript:

1. For training a CNN classifier given the large-size of whole sliding image, the authors implemented a unified memory (UM) CNN. However, this part is not clearly elaborated given the underlying context. In addition, how to deal the large reception field size when the pooling layers are only a few? It's better to give detailed memory consumption and feature map sizes for easier understanding.

2. The authors claimed this is the first study on WSI using only slide-level annotations. This may be incorrect. Following related work also employed similar strategies. The authors should consider comparing and analyzing the differences:

Clinical-grade computational pathology using weakly supervised deep learning on whole slide images. Nature Medicine.

Detection of prostate cancer in whole-slide images through end-to-end training with image-level labels. arXiv preprint.

Weakly Supervised Deep Learning for Whole Slide Lung Cancer Image Analysis. IEEE Transactions on Cybernetics.

3. In the section Baseline MIL Method on Lung Cancer Type Classification, why set $K=1$ for picking up the representative instances?

4. In the paper, the authors analyze the pros and cons for the whole image and MIL based method. For the MIL methods, the comparison is only partial. For example, results from following related literature could be considered for comparison:

Patch-Based Convolutional Neural Network for Whole Slide Tissue Image Classification. CVPR 2016.

Overall, although decent performance has been achieved on the collection data sets, the novelty of the application and methodological parts are limited.

Reviewer #2:

Remarks to the Author:

Histopathology images are very large (tens to hundreds thousands of pixels wide), making it difficult to process at once using usual deep-learning architectures. Often, those images are tiled into smaller images (224x224 pixels for example) to fit into the memory and architectures used.

Major claim

The authors propose a deep-learning method that can deal with very large images. Their technique could therefore offer a novel approach compared to the "tiling" approach usually used in histopathology where the slides are tiled into smaller images treated and trained independently.

Novelty/interest and concerns:

I think there would be an interest in the field of histopathology to use an architecture capable of dealing with large images in an efficient way and this paper is definitely interesting to me. It could potentially be extended to other fields dealing with memory limitations and large images. However, the method and architecture would need to be clarified a lot, and explained better as I had many unanswered technical questions when reading the manuscript. Furthermore, it seems that the technique still requires very powerful GPUs rarely available in most institutions, potentially limiting its usage in the near future. It is however not clear if such requirements are needed because of the architecture itself, or the size of the slides, making it difficult to predict if it's a hard limit or can be alleviated somehow.

Method

1 - Why were TCGA images not used as external validation?

It would be useful to see how it performs on that dataset after having been trained on images from your 3 institutes (both on tissue and diagnostic slides).

2 (minor)- Table 1 describes the number of tiles in each cohort, which is nice.

- could you add more descriptions regarding these 3 input cohorts? Like number of slides in each subtype, size distribution, distribution of tumor cell content, etc...

- for the training done with tiles, can you also specify how many tiles were in each training/validation/test dataset?

3- How is the method performing when compared to the tumor content? Does it perform as well on smaller tumors? Does it require a minimum tumor size for each slide?

4- Figure 2: maybe clarify what the training inputs were (adeno vs other, I guess, with "others" being squamous + normal). Also, it is not clear, did you run 2 binary classifiers, one for adeno vs others, one for squamous vs others, or a 3-way classifier (adeno vs squamous vs normal)? Also, what about the performance on normal tiles?

5- "Experiment results demonstrate a superior performance of our proposed method that achieved AUC scores of 0.950 and 0.924 of adenocarcinoma and squamous cell carcinoma respectively, which outperforms the MIL."

The AUC is indeed superior to the MIL implemented in that paper, but in the literature, there are a few deep-learning architectures used for that purpose that also achieved AUC above 0.9 for lung cancer classification. Those should be mentioned, compared and discussed, and the work should be put in context of the existing literature better.

6- Both methods were done at 4x. Would it be possible to run your method at 20x or even 40x, or would we reach some limitations or memory issues?

7 - Page 5:" For whole-slide training methods, the size of inputs was 20,000 x 20,000 pixels on average." This is confusing because I thought you were taking slides images as they were, whatever their initial sizes. So

7a- What is the distribution of sizes of the slides (specify if it is the size at 4x, or at the magnification at which they were scanned, likely 20x)?

7b- What the max and min sizes that were tested.

7c- Also, slides all have different sizes, so how does the network deals with input of various sizes?

8- What are the parameters / options that were optimized and how were they optimized? I guess like in other network, you must have some learning rate to set, or had to adjust some critical parameters or options in the architecture, but nothing is said about it. The architecture itself is not described enough.

9- In page 7, it says " To control the baseline among training methods and various speed improvement skills, all slides are resized to 10,000 pixels in both width and height dimensions beforehand."

Was it just for that test, or is it the way the input of your architecture is? If all slides are always all resized to 10,000 pixels, then I am confused for 2 reasons:

- first resize to 10k pixels in both sizes would distort images that are not square, right?

- Second, it means that the current architecture is designed to accept larger images than other common networks, but technically, it is still not large enough to contain most of the slides which commonly exceed 10k pixels (this doesn't make your paper less interesting but affect the way it is described!)

10- Page 12: "To be more specific, the MIL method can be separated into two alternative steps: instance selection and classifier optimization, as illustrated in Figure 1."

I don't see how fig 1 illustrates this. Is the figure missing or was it supposed to refer to another one?

11- Visualization:

I appreciate inclusion of the heatmaps, but some clarifications would help the reader:

9a/ For MIL, as it is explained, it sounds like each tile has a probability, and all the pixels of that tile are assigned that probability of being cancerous. If the heatmap was indeed derived this way, and the tiles were non-overlapping as mentioned earlier in the manuscript, why is the heatmap so smooth? One would expect to see the effect of the tiling and "squares" of homogeneous probability
9b/ " Furthermore, our method revealed a more comprehensive ability to highlight all suspicious areas on the slide, especially small lesions." It is true but it also looks like the two heatmaps were not generated the same way (one using CAM, the other using the probability of the tiles), so one could also argue that it would be good to temper such conclusions. In this sentence, it must be clear that the comparison is "our architecture + CAM" vs "MIL + probability heatmap" and not "our architecture" vs "MIL".

12 (minor) "Compared to Coudray et al.¹⁴, training patch-level models with 1,634 slides and tens of millions of patch-level annotations, and Campanella et al.²⁴, training MIL models with over tens of thousands of slides, our model had already achieved a competitive performance with only 7,003 weakly-labeled slides."

Out of curiosity though, if you were to use the same number of slides (1,634), what would be the performance? Or, more generally, did you do a power analysis to see how the number of slides used affect the performance and the minimum required to achieve such a nice AUC?

Code and accessibility:

13 (minor) hardware requirements: "32 GB GPU memory"

I believe many labs don't have access to GPUs with such a large on-card memory (the 32GB are on-card memory, right? - that would limit the scope and potential users). Is there a way to reduce this requirement? For example, by "cropping" slides to make sure they don't exceed a certain size and will fit the memory? Or is it memory independent on the slide size and related to the architecture itself?

14- The README.md file must be **much** more detailed to ensure people can use it independently and list of inputs and parameters/options detailed.

15 (minor) - In the initialization_data.py, it is said that slides without labels are accepted and will be assigned as "-1". I don't see anything in the manuscript that explains how such images would be handled. Or it is outside of the scope of that manuscript?

16- other minor suggestions:

- Page 3: "The main idea of MIL on slide-level cancer classification is that if the patches with highest scores (the most possible K patches) on the slide were identified as carcinoma, the slide should be classified into cancer, and vice versa."

I don't understand what the authors mean by "and vice versa" in this context

- "Moreover, recent studies show that even state-of-the-art weak supervision methods still cannot attain the average performance of strong supervision methods in most computer vision fields such as object detection, semantic segmentation, and instance segmentation tasks."

You should cite appropriate papers when making such statements

- Page 13 - method: please better introduce and explain the principle behind UM mechanisms
- Figure 5: - y axis is a bit confusing: 0.001 images/ sec <> does that mean that when it reads "50", it actually means 50,000 images? It may be more clear to have a 10^3 or $e-3$ beside the

numbers or at the top of the y-axis?

- Also, Fig 5 shows error bars. What are their meaning and if there were repeated experiments, how many repeats?
- Fig8A: annotate explicitly where the GAP layer is
- a more precise diagram of the new architecture would help (with number of layers, sizes of inputs/outputs, etc...)

Reviewer #1 (Remarks to the Author):

This paper presents a deep learning method for whole-slide pathological classification of lung cancer types by using image-level labels only. Experimental results demonstrate good performance on the collected datasets.

However, I have following suggestions to further improve the manuscript:

1. For training a CNN classifier given the large-size of whole sliding image, the authors implemented a unified memory (UM) CNN. However, this part is not clearly elaborated given the underlying context. In addition, how to deal the large reception field size when the pooling layers are only a few? It's better to give detailed memory consumption and feature map sizes for easier understanding.

To better elaborate the implementation detail of UM CNN, we add additional descriptions in the section “**Whole-slide Training Method**” along with a section to describe the proposed optimization techniques in the “**Performance Optimization of Whole-slide Training Method**”. Leveraging unified memory grants hundreds-of-gigabytes-terabyte memory space for GPUs. Therefore, our proposed method allows huge images to be directly trainable for most state-of-the-art CNN architectures, e.g. ResNet, rather than proposing a dedicated model architecture or pipeline, like MIL. To demonstrate the effectiveness of direct training, we adopted ResNet-50 in all experiments except modifying the input size and the aggregation method of the global pooling layer. Besides, we measure the memory consumption of our method and show the results in the section “**Throughput Comparison and Memory Consumption**”.

As for the issue of receptive field size, the receptive field of ResNet-50 is actually large enough to include the cancerous patterns to identify lung cancer types. Modifying ResNet-50 is not required in our case. Please refer to the supplemented descriptions in the section “**MIL Assumption in the Whole-slide Training**”.

2. The authors claimed this is the first study on WSI using only slide-level annotations. This may be incorrect. Following related work also employed similar strategies. The authors should consider comparing and analyzing the differences:

Clinical-grade computational pathology using weakly supervised deep learning on whole slide images. Nature Medicine.

Detection of prostate cancer in whole-slide images through end-to-end training with image-level labels. arXiv preprint.

Weakly Supervised Deep Learning for Whole Slide Lung Cancer Image Analysis. IEEE Transactions on Cybernetics.

Thanks for providing these related works for our method to compare with. We have rewritten the misleading claim in Introduction, meant to be emphasizing our method is the first study to use Unified Memory for learning from slide-level annotations.

We have added these three literatures into the article for comparison. Due to the algorithmic differences between the methods proposed in the first and the third article and ours, we implemented their approaches (MIL-RNN and CNN-MaxFeat-based RF) to compare the model performances. Interestingly, the second paper shares the same objective with ours, that also enables direct training but adopts a totally different way apart from UM. However, there are some limitations of the method proposed by the second paper, e.g. batch normalization layers are not supported.

3. In the section Baseline MIL Method on Lung Cancer Type Classification, why set $K=1$ for picking up the representative instances?

Finding the optimal k for a specific training task requires exhaustive trials. Most studies implement MIL with $k=1$ to not only avoid the time-consuming process but select the most conservative representation. To compare the performance among our proposed method and MIL methods more comprehensively, we added the experiments of MIL with $k=3$ and $k=5$ for comparison. In our case, the MIL with $k=3$ achieves the best performance, while acquires only limited AUC improvements though.

4. In the paper, the authors analyze the pros and cons for the whole image and MIL based method. For the MIL methods, the comparison is only partial. For example, results from following related literature could be considered for comparison:

Patch-Based Convolutional Neural Network for Whole Slide Tissue Image Classification. CVPR 2016.

The current version of the paper compares a variety of MIL variants, including MIL with different k , EM-MIL proposed by the CVPR paper, MIL-RNN and CNN-MaxFeat-based RF mentioned in Q2. The performances of these methods are shown in the first section “**Model Performance of Multiple Instance Learning (MIL)**” in Result.

Overall, although decent performance has been achieved on the collection data sets, the novelty of the application and methodological parts are limited.

Reviewer #2 (Remarks to the Author):

Histopathology images are very large (tens to hundreds thousands of pixels wide), making it difficult to process at once using usual deep-learning architectures. Often, those images are tiled into smaller images (224x224 pixels for example) to fit into the memory and architectures used.

*** Major claim**

The authors propose a deep-learning method that can deal with very large images. Their technique could therefore offer a novel approach compared to the "tiling" approach usually used in histopathology where the slides are tiled into smaller images treated and trained independently.

*** Novelty/interest and concerns:**

I think there would be an interest in the field of histopathology to use an architecture capable of dealing with large images in an efficient way and this paper is definitely interesting to me. it could potentially be extended to other fields dealing with memory limitations and large images.

However, the method and architecture would need to be clarified a lot, and explained better as I had many unanswered technical questions when reading the manuscript. Furthermore, it seems that the technique still requires very powerful GPUs rarely available in most institutions, potentially limiting its usage in the near future. It is however not clear if such requirements are needed because of the architecture itself, or the size of the slides, making it difficult to predict if it's a hard limit or can be alleviated somehow.

Method

1 - Why were TCGA images not used as external validation?

It would be useful to see how it performs on that dataset after having been trained on images from your 3 institutes (both on tissue and diagnostic slides).

As suggested, we added the testing performance on TCGA diagnostic slides and tissue slides in the section “**Model Performance on TCGA Datasets**”. The results on diagnostic slides were quite positive (AUC > 0.9), demonstrating high generalization ability of the model on slides acquired from different sites. However, inferencing on the TCGA tissue slides only yields 0.6~0.7 AUC scores. One sound reason is its fixation method adopted, frozen section procedure instead of FFPE. Morphological changes are a common artifact when using frozen procedure, which makes the model trained on frozen slides hard to distinguish lung cancer types.

2 (minor)- Table 1 describes the number of tiles in each cohort, which is nice.

- could you add more descriptions regarding these 3 input cohorts? Like number of slides in each subtype, size distribution, distribution of tumor cell content, etc...

- for the training done with tiles, can you also specify how many tiles were in each training/validation/test dataset?

More information of our datasets is reported as long as they are retrievable from the medical records in the current version of the article, including

(1) the maximal and average sizes of the slide images (in **Dataset** section),

(2) the percentage of slides sampled by biopsy or resection (in **Dataset** section),

(3) the fixation method of the slides, FFPE, (in **Dataset** section), and

(4) rough numbers of tiles (in **MIL Methods on Lung Cancer Type Classification** section).

3- How is the method performing when compared to the tumor content? Does it perform as well on smaller tumors? Does it require a minimum tumor size for each slide?

To evaluate whether our models can identify tumor cells given slides with small lesion areas, we further picked out slides which tumor occupies less than 10% of the tissue area from the testing set (described in **Dataset** section). Our method can still achieve $AUC > 0.92$ even for this hard case (described in **Model Performance on Small Lesion Testing Dataset** section).

4- Figure 2: maybe clarify what the training inputs were (adeno vs other, I guess, with "others" being squamous + normal). Also, it is not clear, did you run 2 binary classifiers, one for adeno vs others, one for squamous vs others, or a 3-way classifier (adeno vs squamous vs normal)? Also, what about the performance on normal tiles?

We train 3-way classifiers for most experiments; that is, the models output probabilities of adeno, squamous and normal respectively given a slide input (described in **Results** section). Additionally, we found the performances of separated binary classifiers respectively for adeno and squamous are almost the same as those a single 3-class classifier can achieve (described in **3-class Classifier versus Individual Binary Classifiers** section).

The AUC scores of normal basically measure whether a model can distinguish normal and abnormal (adeno+squamous) slides, which is in fact a simpler task compared to classifying main types. Quantitatively, this score is always higher than the other two, e.g. 0.9679 for our model (vs 0.9594 and 0.9414). We do not report this score since we would like to focus on how models perform on main type classification.

5- “Experiment results demonstrate a superior performance of our proposed method that achieved AUC scores of 0.950 and 0.924 of adenocarcinoma and squamous cell carcinoma respectively, which outperforms the MIL.”

The AUC is indeed superior to the MIL implemented in that paper, but in the literature, there are a few deep-learning architectures used for that purpose that also achieved AUC above 0.9 for

lung cancer classification. Those should be mentioned, compared and discussed, and the work should be put in context of the existing literature better.

We did plenty of experiments on comparing our method against state-of-the-art methods leveraging slide-level labels, including MIL with different k , EM-MIL proposed by Hou et al., MIL-RNN proposed by Campanella et al. and CNN-MaxFeat-based RF proposed by Wang et al.. We hope the substantial revision could support the claim that our method performs better.

6- Both methods were done at 4x. Would it be possible to run your method at 20x or even 40x, or would we reach some limitations or memory issues?

Currently, our method cannot tackle 20x and 40x slide images due to host memory constraint. As shown in Figure 9, the huge amount of intermediate data produced during training on 4x images is hundreds of GBs large, already reaching the current hardware limitation. Seeking approaches to minimize memory consumption remains an active issue for us to further improve the whole-slide training method.

Training MIL on larger images is runnable, but, however, requires large amounts of computing resources and time to reach model convergence. In fact, we have tried training MIL on 8x images (40,000 x 40,000) by 8 GPUs in parallel. In the end, the experiment was terminated since the estimated training period is over one year. It is basically computationally infeasible unless more computing resources are provided.

Despite the fact that we cannot show 4x images are detailed enough, tasks to identify tissue-level morphology like our case, lung cancer type classification, generally do not require higher magnification for human experts.

7 - Page 5:" For whole-slide training methods, the size of inputs was 20,000 x 20,000 pixels on average." This is confusing because I thought you were taking slides images as they were, whatever their initial sizes. So

7a- What is the distribution of sizes of the slides (specify if it is the size at 4x, or at the magnification at which they were scanned, likely 20x)?

7b- What the max and min sizes that were tested.

7c- Also, slides all have different sizes, so how does the network deal with input of various sizes?

The original 20x images were scanned with width up to 110,592 pixels (average: 65,683 pixels) and height up to 55,552 pixels (average: 40,593 pixels), as described in **Dataset** section. These slides were resized to 4x magnification (i.e. 0.2x the original resolution), and then padded to 21,500 x 21,500 pixels to unify image resolution in datasets (described in **Results and Method** section).

8- What are the parameters / options that were optimized and how were they optimized? I guess like in other network, you must have some learning rate to set, or had to adjust some critical parameters or options in the architecture, but nothing is said about it. The architecture itself is not described enough.

Actually, minimizing required efforts on hyperparameter tuning is what we pursue. It is impractical to tune hyperparameters since it requires multiple trials to collect empirical data, which is time-consuming, resource-wasting, and tends to bias to tuning datasets. According to our implementation experiences, methods with many hyperparameter tuning are prone to produce selection bias; that is, the performances of proposed methods are often overestimated due to careful tuning.

Our proposed method introduces no additional hyperparameter, making it plain but useful. The same philosophy also applies to our choice of the underlying model architecture, ResNet, and the optimizer, Adam, which are particularly not sensitive to hyperparameter tuning. As for the lack in describing our method, we added more descriptions in the **Whole-slide Training Method** section and added a new section, **Performance Optimization of Whole-slide Training Method**, to introduce how we speed up the whole process.

9- In page 7, it says " To control the baseline among training methods and various speed improvement skills, all slides are resized to 10,000 pixels in both width and height dimensions beforehand."

Was it just for that test, or is it the way the input of your architecture is? If all slides are always all resized to 10,000 pixels, then I am confused for 2 reasons:

- first resize to 10k pixels in both sizes would distort images that are not square, right?
- Second, it means that the current architecture is designed to accept larger images than other common networks, but technically, it is still not large enough to contain most of the slides which commonly exceed 10k pixels (this doesn't make your paper less interesting but affect the way it is described!)

For throughput tests, we adopt the same image pre-processing method on all experiments. The method is described in the first paragraph of the **Results** section, and also the answer to Q7. By that way, images are not distorted when downsampling and preserve their morphology. As for the second question, yes, supporting larger inputs is bounded by the host memory limitation, as answered in Q6. (By the way, thanks for your interest in our study!)

10- Page 12: "To be more specific, the MIL method can be separated into two alternative steps: instance selection and classifier optimization, as illustrated in Figure 1."

I don't see how fig 1 illustrates this. Is the figure missing or was it supposed to refer to another one?

Please refer to Figure 11(B). We made the figure more informative and consist with how we describe in paragraphs.

11- Visualization:

I appreciate inclusion of the heatmaps, but some clarifications would help the reader:

9a/ For MIL, as it is explained, it sounds like each tile has a probability, and all the pixels of that tile are assigned that probability of being cancerous. If the heatmap was indeed derived this way, and the tiles were non-overlapping as mentioned earlier in the manuscript, why is the heatmap so smooth? One would expect to see the effect of the tiling and "squares" of homogeneous probability

You are correct, while we adopt a different way to visualize the heatmap to make highlights more aligned with lesion boundaries. Basically, both the heatmaps of MIL and our method are originally small images. Before overlaying on their corresponding slide images, they are upsampled to fit the size. When choosing nearest-neighbor interpolation as the upsampling method, the heatmap will be what you described. We adopt bicubic interpolation instead since we assume probabilities of tumor lesions are in round shapes (described in the caption of Figure 4).

9b/ " Furthermore, our method revealed a more comprehensive ability to highlight all suspicious areas on the slide, especially small lesions." It is true but it also looks like the two heatmaps were not generated the same way (one using CAM, the other using the probability of the tiles), so one could also argue that it would be good to temper such conclusions. In this sentence, it must be clear that the comparison is "our architecture + CAM" vs "MIL + probability heatmap" and not "our architecture" vs "MIL".

We agree with you that the better visualization effect of our method + CAM does not mean our method has better ability on identifying small lesions, so we removed that sentence. To quantitatively compare the performances on small lesion slides, we collected a small-lesion testing set and recorded the experimental results in the **Model Performance on Small Lesion Testing Dataset** section.

12 (minor) "Compared to Coudray et al.¹⁴, training patch-level models with 1,634 slides and tens of millions of patch-level annotations, and Campanella et al.²⁴, training MIL models with over tens of thousands of slides, our model had already achieved a competitive performance with only 7,003 weakly-labeled slides."

Out of curiosity though, if you were to use the same number of slides (1,634), what would be the performance? Or, more generally, did you do a power analysis to see how the number of slides used affect the performance and the minimum required to achieve such a nice AUC?

Please see the **Impact of Dataset Size and Image Resolution** section for more information about statistical analysis on dataset-size/image-resolution effect. Besides measuring

the impact of dataset reduction, we also discussed the influence of less image resolution (2x and 1x magnifications). As the results shown in Figure 7, AUCs of model constantly decrease as less and less training data are given.

Code and accessibility:

13 (minor) hardware requirements: "32 GB GPU memory"

I believe many labs don't have access to GPUs with such a large on-card memory (the 32GB are on-card memory, right? - that would limit the scope and potential users). Is there a way to reduce this requirement? For example, by "cropping" slides to make sure they don't exceed a certain size and will fit the memory? Or is it memory independent on the slide size and related to the architecture itself?

Please ignore that requirement. In fact, sufficient host memory is instead the hardware requirement, as described in the **Throughput Comparison and Memory Consumption** section. For computers with limited memory resources, a good starter is to try smaller images, e.g. 10,000 x 10,000. We showed the performance of less resolution is still acceptable in the **Impact of Dataset Size and Image Resolution** section, at least for our case. Another way is like what you suggested, cropping regions of interests (RoIs) out of a slide, doing RoI-level annotations and training models with RoI images.

14- The README.md file must be *much* more detailed to ensure people can use it independently and list of inputs and parameters/options detailed.

Exactly, so we rewrite the whole program as well as the README file. We hope the revised version is clear enough.

15 (minor) - In the initialization_data.py, it is said that slides without labels are accepted and will be assigned as "-1". I don't see anything in the manuscript that explains how such images would be handled. Or it is outside of the scope of that manuscript?

This is no longer an issue in the new program.

16- other minor suggestions:

- Page 3: “The main idea of MIL on slide-level cancer classification is that if the patches with highest scores (the most possible K patches) on the slide were identified as carcinoma, the slide should be classified into cancer, and vice versa.”

I don't understand what the authors mean by “and vice versa” in this context

That means if patches in a slide with highest likelihoods to be cancerous are identified as normal tissues, the slide should be normal. We have revised it in the **Introduction** section.

- “Moreover, recent studies show that even state-of-the-art weak supervision methods still cannot attain the average performance of strong supervision methods in most computer vision fields such as object detection, semantic segmentation, and instance segmentation tasks.”

You should cite appropriate papers when making such statements

3 citations have been added to support that statement, which are state-of-the-art papers proposing weakly supervised training methods.

- Page 13 - method: please better introduce and explain the principle behind UM mechanisms

Please see the supplemented descriptions in the **Whole-slide Training Method** and the new section, **Performance Optimization of Whole-slide Training Method**. The former one introduces how the model extends to support large image inputs. The latter one explains the detailed mechanism of UM and how we optimize it.

- Figure 5: - y axis is a bit confusing: 0.001 images/ sec \Leftrightarrow does that mean that when it reads "50", it actually means 50,000 images? It may be more clear to have a 10^3 or $e-3$ beside the numbers or at the top of the y-axis?

We adopted images per “minutes” now to prevent confusion.

- Also, Fig 5 shows error bars. What are their meaning and if there were repeated experiments, how many repeats?

Yes, we recorded the elapsed times of 30 training iterations. The error bars represent 95% confidence intervals of sampled elapsed times (described in the caption of Figure 8).

- Fig8A: annotate explicitly where the GAP layer is

We replaced “GMP layer” by “GAP/GMP layer” in the figure.

- a more precise diagram of the new architecture would help (with number of layers, sizes of inputs/outputs, etc...)

As described in the **Whole-slide Training Method**, ResNet-50 we adopted in the experiments stays unmodified except for the input size and the global pooling layer.

Reviewers' Comments:

Reviewer #1:

Remarks to the Author:

I appreciate the efforts from authors for trying to address my concerns. Although decent performance has been achieved on the collection data sets, the novelty of the application and methodological parts compared to the existing study are limited. The performance justification by leveraging unified memory is also unclear. In addition, more challenging part on whole slide image analysis in 40x and 20x remains to be resolved.

Reviewer #2:

Remarks to the Author:

The authors have addressed all questions and suggestions in a way that, in my opinion, makes the paper stronger.

I have no more comment.

Reviewer #1 (Remarks to the Author):

I appreciate the efforts from authors for trying to address my concerns. Although decent performance has been achieved on the collection data sets, the novelty of the application and methodological parts compared to the existing study are limited. The performance justification by leveraging unified memory is also unclear. In addition, more challenging part on whole slide image analysis in 40x and 20x remains to be resolved.

Thank you for your valuable comment. We would like to point out that compared with existing work, our research represents significant progress, for the following three reasons. 1) The scale of our research using intact whole slide images for deep learning is of the largest scale so far, whether it's in slide number (7003 vs 1243), pixel resolution (21500 x 21500 vs 16384 x 16384), or number of experimental conditions. 2) The large scale of our research enabled us to clearly demonstrate that training deep neural network using intact whole slide images results in superior classification performance over previously established method for weakly supervised learning (i.e., multiple instance learning), especially when the classification task is more difficult (ternary vs binary classification) or when the tissue area is very large. 3) Our work demonstrates the effectiveness of deep neural networks trained on intact whole slide images in recognizing small lesions. In summary, our work clearly demonstrates the usefulness of using intact whole slide images to train deep neural networks for image recognition tasks and points out important directions for future research.

Indeed, the challenge with image recognition tasks at 20x or 40x remains. We believe this problem need not be solved solely by increasing the dimension of the input images. This can be solved by breaking the image recognition problem into steps. For example, it is possible to take a step-wise approach, first using a 4x image to locate regions of interest, then using images of these regions at 40x for the final image recognition task. This would in fact resemble the behavior of a pathologist in real life.

Reviewer #2 (Remarks to the Author):

The authors have addressed all questions and suggestions in a way that, in my opinion, makes the paper stronger.

I have no more comment.

Thank you for the comments to provide guidance on how we could improve the paper. We also think our work has gained more robustness after the reviewers' inputs, and feel delighted we could answer your questions.